# A Decoupled Doppler Positioning Algorithm for Dynamic Receivers Using LEO Constellation Signals

**DOI:** 10.3390/s25216760

**Published:** 2025-11-05

**Authors:** Tianqi Liu, Yan Liu, Chenggan Wen, Yonghang Jiang, Linxiong Wang, Rong Yang, Jiong Yi

**Affiliations:** 1School of Physics and Microelectronics, Hunan University, Changsha 410082, China; liutianqi@hnu.edu.cn; 2College of Semiconductors (College of Integrated Circuits), Changsha Semiconductor Technology and Application Innovation Research Institute, Hunan University, Changsha 410006, China; wenchenggan@hnu.edu.cn (C.W.); 19330899095@163.com (L.W.); 3Changsha Jinwei Integrated Circuit Co., Ltd., Changsha 410205, China; liuyan@cs-jinwei.com; 4College of Computer Science and Technology, National University of Defense Technology, Changsha 410073, China

**Keywords:** dynamic Doppler positioning, LEO satellites, LS method, zero initial value, velocity measurement

## Abstract

The advancement of low-earth-orbit (LEO) communication constellations has revitalized interest in Doppler-based positioning. However, conventional Doppler positioning algorithms struggle with dynamic receivers under unknown initial states due to the inherent nonlinearity of the observation model. To address this challenge, we propose an improved least-squares-based algorithm that decouples the estimation of position and velocity, enabling robust positioning from a zero initial state. Simulation results demonstrate that the proposed method achieves meter-level positioning accuracy and decimeter-per-second velocity accuracy under various dynamic scenarios, including high-speed motion. This approach establishes a viable framework for real-time navigation in GNSS-challenged environments using LEO signals.

## 1. Introduction

Artificial LEO satellites are crucial for global communication, environmental monitoring, and global positioning and navigation. The advancement of emerging technologies such as Inter-Satellite Link (ISL) and navigation enhancement has strengthened satellite system independence and transmission efficiency, leading to renewed growth in LEO satellite constellation development [1,2,3,4,5]. Globally, the deployment of LEO constellations is accelerating, with projects like the Starlink in the United States [6] and OneWeb in the United Kingdom [7,8] launching thousands of satellites. As of August 2025, the number of in-orbit LEO satellites has exceeded 10,000, solidifying LEO-based technologies as a primary focus in navigation research.

LEO satellites support a wide variety of applications beyond communications, including Earth observation, weather monitoring, and Positioning, Navigation, and Timing (PNT) services. In the field of PNT, LEO constellations can enhance traditional GNSS systems by providing stronger signals, faster convergence, and improved robustness in challenging environments [9,10,11,12,13]. Moreover, they can independently provide PNT services. Although the onboard clocks—typically disciplined oscillators rather than atomic clocks—constrain the achievable time accuracy [14], the high orbital dynamics and signal strength of LEO satellites make Doppler positioning a particularly advantageous and prominent method for LEO-based navigation.

The concept of Doppler positioning is well-established, as demonstrated by the historic Transit system, which achieved positioning by utilizing Doppler shifts [15]. With the advent of dense LEO constellations, multi-satellite Doppler positioning has become feasible. Systems like Iridium have pioneered this application, using 66 satellites to provide services, with researchers often leveraging their opportunity signals for positioning [16]. The core algorithm for multi-satellite Doppler positioning is the Least-Squares (LS) method. For static receivers, this method involves only four unknowns and has achieved positioning accuracy of around 22 m using real signals [17,18], which can be further improved to about 10 m with refined Doppler shift deviation modeling [19]. For dynamic receivers, the LS equation expands to seven unknowns (position, velocity, and clock offset rate), and sub-meter positioning accuracy has been reported in simulation scenarios [20,21]. Besides Doppler-based methods, recent studies have explored Angle-of-Arrival (AoA) estimation using LEO signals of opportunity for coarse positioning [22,23]. While AoA approaches offer complementary benefits, this work focuses on Doppler-based positioning due to its inherent robustness to frequency offsets and compatibility with existing communication infrastructure. However, a critical limitation remains: the positioning result is highly sensitive to the initial value of the receivers, especially for low orbital satellites. Studies indicate that the initial-to-actual value error must remain below 300 km to avoid significant deviations [15]. This poses a severe challenge for dynamic receivers with unknown initial positions, such as during the first start-up, where the initial position error could be as large as 20,000 km. Although methods like Tikhonov regularization have been proposed to estimate the initial position [24], most existing research focuses on static or low-speed scenarios, and a robust solution for truly dynamic receivers starting from an unknown state is still lacking.

The core of this challenge lies in the inherent nonlinearity of the Doppler positioning equation, where a multiplicative relationship exists between the receiver’s position and velocity unknowns. This introduces quadratic terms that complicate the linearization process. To fundamentally address this issue, our work introduces an improved LS-based algorithm that innovatively separates the calculation of position and velocity. This decoupling strategy simplifies the equation linearization process. Our algorithm enables the positioning of dynamic objects starting from an initial value of zero, delivering meter-level position accuracy and decimeter-per-second-level velocity accuracy, thereby overcoming the primary convergence barrier of traditional methods.

The remainder of this paper is organized as follows: Section 2 details the theory and the proposed algorithm. Section 3 describes the simulation data sources. Section 4, Section 5, Section 6 and Section 7 present extensive experimental results and analysis under various conditions. Finally, Section 8 concludes the paper.

## 2. Theory an Algorithm

### 2.1. Doppler Positioning Model

Due to the relative motion between a satellite and a receiver, the frequency of a signal transmitted from the satellite changes when it is received. This frequency difference, known as the Doppler shift, is denoted as $df=f-f_{0}$, where f is the receiving frequency and $f_{0}$ is the nominal frequency. The Doppler shift df can be modeled by Equation (1):
(1)$$df=\frac{f_{0}}{c}\cdot \left(v^{s}-v\right)\cdot \frac{r^{s}-r}{\left\|r^{s}-r\right\|}+f_{u}+\varepsilon_{f},$$
where c is the speed of light. $v^{s}={\left({v_{x}}^{s},{v_{y}}^{s},{v_{z}}^{s}\right)}^{T}$ is the satellite velocity vector. $v={\left(v_{x},v_{y},v_{z}\right)}^{T}$ is the receiver velocity vector. $r^{s}={\left(x^{s},y^{s},z^{s}\right)}^{T}$ denotes the satellite position vector. $r={\left(x,y,z\right)}^{T}$ denotes the receiver position vector. $f_{u}$ is the offset rate of the receiver clock. $\varepsilon_{f}$ is the random noise.

In Equation (1), f is easily observed. $f_{0}$ is a definite value. c is a constant. $v^{s}$ and $r^{s}$ can be calculated using ephemeris. The remaining parameters r, v and $f_{u}$ are unknown, resulting in a total of seven unknowns. With a sufficient number of satellite observations, the system of equations can be solved using the Least-Squares (LS) method to determine the receiver’s position and velocity. The system of equations for *n* satellites (*n* ≥ 7) is given by
(2)$$\left\{\begin{array}{c} {df}^{1}=\frac{f_{0}}{c}\cdot \left(v_{1}^{s}-v\right)\cdot \frac{r_{1}^{s}-r}{\left\|r_{1}^{s}-r\right\|}+f_{u}+{\varepsilon_{f}}_{1}\\ \vdots \\ {df}^{i}=\frac{f_{0}}{c}\cdot \left(v_{i}^{s}-v\right)\cdot \frac{r_{i}^{s}-r}{\left\|r_{i}^{s}-r\right\|}+f_{u}+{\varepsilon_{f}}_{i}\\ \vdots \\ {df}^{n}=\frac{f_{0}}{c}\cdot \left(v_{n}^{s}-v\right)\cdot \frac{r_{n}^{s}-r}{\left\|r_{n}^{s}-r\right\|}+f_{u}+{\varepsilon_{f}}_{n} \end{array}\right.,$$
where the superscript represents the satellite index.

### 2.2. Traditional Method

The traditional algorithm requires an initial guess of the receiver state. The system of Equation (2) is linearized around this initial point using a first-order Taylor expansion. The initial point state vector is represented by $X_{0}={\left[\begin{array}{ccc} r_{0} & v_{0} & {f_{u}}_{0} \end{array}\right]}^{T}={\left[\begin{array}{ccccccc} x_{0} & y_{0} & z_{0} & {v_{x}}_{0} & {v_{y}}_{0} & {v_{z}}_{0} & f_{u_{0}} \end{array}\right]}^{T}$. The true value of the state vector was $X={\left[\begin{array}{ccc} r & v & f_{u} \end{array}\right]}^{T}={\left[\begin{array}{ccccccc} x & y & z & v_{x} & v_{y} & v_{z} & f_{u} \end{array}\right]}^{T}$. The linearized system can be expressed in matrix form as
(3)F=G·dX,
where the update vectors are $dX=X-X_{0}$. The Doppler rate observation vector $F={\left[\delta \begin{array}{ccccc} {df}^{1} & \cdots & \delta {df}^{i} & \cdots & {\delta df}^{n} \end{array}\right]}^{T}$. In this form of expression, $\delta {df}^{i}={df}^{i}-{df}^{i}(X_{0})$, where ${df}^{i}(X_{0})$ is calculated using Formula (1) when $X=X_{0}$. The parameter matrix $G=\left[\begin{array}{ccccccc} G_{x}^{1} & G_{y}^{1} & G_{z}^{1} & G_{vx}^{1} & G_{vy}^{1} & G_{vz}^{1} & 1\\ \vdots & \vdots & \vdots & \vdots & \vdots & \vdots & \vdots \\ G_{x}^{i} & G_{y}^{i} & G_{z}^{i} & G_{vx}^{i} & G_{vy}^{i} & G_{vz}^{i} & 1\\ \vdots & \vdots & \vdots & \vdots & \vdots & \vdots & \vdots \\ G_{x}^{n} & G_{y}^{n} & G_{z}^{n} & G_{vx}^{n} & G_{vy}^{n} & G_{vz}^{n} & 1 \end{array}\right]$, in which
(4)$$\left\{\begin{array}{c} G_{x}^{i}=\frac{f_{0}}{c}\cdot \frac{{{v_{x}}^{s}}_{i}-v_{x_{0}}}{\left\|r_{0}-r_{i}^{s}\right\|}-\frac{f_{0}}{c}\cdot \left(x_{0}-x_{i}^{s}\right)\cdot \frac{(v_{i}^{s}-v_{0})\cdot \left(r_{0}-r_{i}^{s}\right)}{{\left(\left\|r_{0}-r_{i}^{s}\right\|\right)}^{3}}\\ G_{y}^{i}=\frac{f_{0}}{c}\cdot \frac{{{v_{y}}^{s}}_{i}-v_{y_{0}}}{\left\|r_{0}-r_{i}^{s}\right\|}-\frac{f_{0}}{c}\cdot \left(y_{0}-y_{i}^{s}\right)\cdot \frac{(v_{i}^{s}-v_{0})\cdot \left(r_{0}-r_{i}^{s}\right)}{{\left(\left\|r_{0}-r_{i}^{s}\right\|\right)}^{3}}\\ G_{z}^{i}=\frac{f_{0}}{c}\cdot \frac{{{v_{z}}^{s}}_{i}-v_{z_{0}}}{\left\|r_{0}-r_{i}^{s}\right\|}-\frac{f_{0}}{c}\cdot \left(z_{0}-z_{i}^{s}\right)\cdot \frac{(v_{i}^{s}-v_{0})\cdot \left(r_{0}-r_{i}^{s}\right)}{{\left(\left\|r_{0}-r_{i}^{s}\right\|\right)}^{3}}\\ G_{vx}^{i}=\frac{f_{0}}{c}\cdot \frac{-\left(x_{0}-x_{i}^{s}\right)}{\left\|r_{0}-r_{i}^{s}\right\|}\\ G_{vy}^{i}=\frac{f_{0}}{c}\cdot \frac{-\left(y_{0}-y_{i}^{s}\right)}{\left\|r_{0}-r_{i}^{s}\right\|}\\ G_{vz}^{i}=\frac{f_{0}}{c}\cdot \frac{-\left(z_{0}-z_{i}^{s}\right)}{\left\|r_{0}-r_{i}^{s}\right\|}\\ 1 \end{array}\right.,$$

The state update dX is then calculated using the Least-Squares solution:
(5)$$dX={\left(G^{T}G\right)}^{-1}G^{T}F,$$
(6)$$X=X_{0}+dX,$$

This process repeats until the magnitude of dX falls below a predefined threshold (such as $10^{-6}$). If the threshold is not met, set $X_{0}=X$. Then, repeat the calculation of Equations (5) and (6). If so, set the final result X as the final result. Thus, the position and velocity of the receiver can be obtained. A key limitation of this method is that the initial state vector $X_{0}$ is often set to zero, which can lead to convergence problem due to the second-order coupling between receiver position and velocity in the observation model.

### 2.3. Decoupled Method

To overcome the limitations of the traditional approach, we propose an algorithm that decouples the position and velocity estimation. This separation eliminates the influence of second-order coupling terms, thereby enhancing convergence stability. First, assuming that the position of receiver was r=0, only remaining unknowns are state vectors v and $f_{u}$. At this point, state vector is $V={\left[\begin{array}{cc} v & f_{u} \end{array}\right]}^{T}={\left[\begin{array}{cccc} v_{x} & v_{y} & v_{z} & f_{u} \end{array}\right]}^{T}$. At the initial point, state vector was $V_{0}={\left[\begin{array}{cc} v_{0} & f_{u_{0}} \end{array}\right]}^{T}={\left[\begin{array}{cccc} {v_{x}}_{0} & {v_{y}}_{0} & {v_{z}}_{0} & f_{u_{0}} \end{array}\right]}^{T}$. Then, the system of equations can be written in matrix form. In this section, the matrix equation is as follows:
(7)$$F_{1}=G_{1}\cdot dV,$$
where the update vectors are $dV=V-V_{0}$. The Doppler rate observation vector $F_{1}={\left[\delta \begin{array}{ccccc} {df}^{1} & \cdots & \delta {df}^{i} & \cdots & {\delta df}^{n} \end{array}\right]}^{T}$. In this form of expression, $\delta {df}^{i}={df}^{i}-{df}^{i}(V_{0})$, in which ${df}^{i}(V_{0})$ is calculated using Formula (1) when $V=V_{0}$ (r is known). The parameter matrix $G_{1}=\left[\begin{array}{cccc} G_{vx}^{1} & G_{vy}^{1} & G_{vz}^{1} & 1\\ \vdots & \vdots & \vdots & \vdots \\ G_{vx}^{i} & G_{vy}^{i} & G_{vz}^{i} & 1\\ \vdots & \vdots & \vdots & \vdots \\ G_{vx}^{n} & G_{vy}^{n} & G_{vz}^{n} & 1 \end{array}\right]$, in which
(8)$$\left\{\begin{array}{c} G_{vx}^{i}=\frac{f_{0}}{c}\cdot \frac{-\left(x-x_{i}^{s}\right)}{\left\|\begin{array}{c} r_{0}-r_{i}^{s} \end{array}\right\|}\\ G_{vy}^{i}=\frac{f_{0}}{c}\cdot \frac{-\left(y-y_{i}^{s}\right)}{\left\|\begin{array}{c} r_{0}-r_{i}^{s} \end{array}\right\|}\\ G_{vz}^{i}=\frac{f_{0}}{c}\cdot \frac{-\left(z-z_{i}^{s}\right)}{\left\|\begin{array}{c} r_{0}-r_{i}^{s} \end{array}\right\|} \end{array}\right.,$$

In general, the initial vector $V_{0}$ is set to 0. Then, dV can be calculated as
(9)$$dV={\left({G_{1}}^{T}G_{1}\right)}^{-1}{G_{1}}^{T}F_{1},$$
(10)$$V=dV+V_{0},$$

In this way, the receiver′s velocity was obtained.

Then, update the value of the receiver position. The state vector was $R={\left[\begin{array}{cc} r & f_{u} \end{array}\right]}^{T}={\left[\begin{array}{cccc} x & y & z & f_{u} \end{array}\right]}^{T}$. Unfold the system of equations at the expansion point (The position of the point is $R_{0}={\left[\begin{array}{cc} r_{0} & {f_{u_{0}}}^{\prime } \end{array}\right]}^{T}={\left[\begin{array}{cccc} x_{0} & y_{0} & z_{0} & {f_{u_{0}}}^{\prime } \end{array}\right]}^{T}$. The values of velocity and ${f_{u_{0}}}^{\prime }$ originate from V. Equation (11) was obtained using the LS method:
(11)$$F_{2}=G_{2}\cdot dR,$$

In this equation, $dR=R-R_{0}$. $F_{2}={\left[\delta \begin{array}{ccccc} {{df}^{1}}^{\prime } & \cdots & {\delta {df}^{i}}^{\prime } & \cdots & {\delta {df}^{n}}^{\prime } \end{array}\right]}^{T}$, in which ${\delta {df}^{i}}^{\prime }={df}^{i}-{df}^{i}(R_{0})$, in which ${df}^{i}(R_{0})$ is calculated by Formula (1) when $R=R_{0}$ (the velocity of the receiver is known). $G_{2}$ matrix is expressed as $\left[\begin{array}{cccc} G_{x}^{1} & G_{y}^{1} & G_{z}^{1} & 1\\ \vdots & \vdots & \vdots & \vdots \\ G_{x}^{i} & G_{y}^{i} & G_{z}^{i} & 1\\ \vdots & \vdots & \vdots & \vdots \\ G_{x}^{n} & G_{y}^{n} & G_{z}^{n} & 1 \end{array}\right]$. The parameters were calculated as follows:
(12)$$\left\{\begin{array}{c} G_{x}^{i}=\frac{f_{0}}{c}\cdot \frac{{{v_{x}}^{s}}_{i}-v_{x_{1}}}{\left\|r_{0}-r_{i}^{s}\right\|}-\frac{f_{0}}{c}\cdot \left(x_{0}-x_{i}^{s}\right)\cdot \frac{(v_{i}^{s}-v_{1})\cdot \left(r_{0}-r_{i}^{s}\right)}{{\left(\left\|r_{0}-r_{i}^{s}\right\|\right)}^{3}}\\ G_{y}^{i}=\frac{f_{0}}{c}\cdot \frac{{{v_{y}}^{s}}_{i}-v_{y_{1}}}{\left\|r_{0}-r_{i}^{s}\right\|}-\frac{f_{0}}{c}\cdot \left(y_{0}-y_{i}^{s}\right)\cdot \frac{(v_{i}^{s}-v_{1})\cdot \left(r_{0}-r_{i}^{s}\right)}{{\left(\left\|r_{0}-r_{i}^{s}\right\|\right)}^{3}}\\ G_{z}^{i}=\frac{f_{0}}{c}\cdot \frac{{{v_{z}}^{s}}_{i}-v_{z_{1}}}{\left\|r_{0}-r_{i}^{s}\right\|}-\frac{f_{0}}{c}\cdot \left(z_{0}-z_{i}^{s}\right)\cdot \frac{(v_{i}^{s}-v_{1})\cdot \left(r_{0}-r_{i}^{s}\right)}{{\left(\left\|r_{0}-r_{i}^{s}\right\|\right)}^{3}} \end{array}\right.,$$

dR was calculated as
(13)$$dR={\left({G_{2}}^{T}G_{2}\right)}^{-1}{G_{2}}^{T}F_{2},$$
(14)$$R=dR+R_{0},$$

The proposed method employs the displacement vector $dr={\left[\begin{array}{ccc} x-x_{0} & y-x_{0} & z-x_{0} \end{array}\right]}^{T}$ as the convergence criterion for terminating the iterative process. A threshold value k (e.g., $10^{-6}$) is introduced to quantify convergence. If the value of $\left\|dr\right\|$ is less than the threshold value, the iteration terminates and the solution is deemed converged. When the value of $\left\|dr\right\|$ is larger than k, two distinct iterative strategies are implemented based on the state of the receiver.

During the receiver start-up state, where substantial positional error may exist, a position-only iteration strategy is adopted. Specifically, in every second, the initial reference position $R_{0}$ is first iteratively updated to the current position estimate R, followed by recalculation of the system parameters through Equations (13) and (14). This procedure continues until the norm condition $\left\|dr\right\|\leq k$ is satisfied. The number of algorithm iterations in this process is often greater than 10. During the start-up state, this process typically lasts for several tens of seconds.

When the number of iterations was less than 10, the update of parameters is carried out in this form: $R_{0}=R$ and $V_{0}=V$. After this preconditioning state, the system iteratively solves Equations (9), (10), (13), and (14) until the convergence criterion $\left\|dr\right\|\leq k$ is satisfied. The above algorithm is illustrated in the workflow (Figure 1).

The core innovation of this proposed algorithm lies in its adaptive velocity iteration control. When the position is not precise, the velocity is calculated only once and each iteration only calculates the position. After the algorithm stabilized, the speed and position were calculated once per iteration. This phased approach strategically allocates computational resources, prioritizing position stabilization before engaging full parameter optimization to enhance convergence efficiency.

## 3. Simulation Data Sources

Owing to the limited public availability of real LEO satellite signals and dynamic observation data, this study employed the GW constellation simulator (XSS6000) to investigate the Doppler positioning problem in dynamic scenarios. The simulation data was provided by Hunan Weidao Information Technology Co., Ltd., Changsha, China.

The GW constellation is a Chinese LEO satellite system currently under construction. To date, 116 satellites have been launched, with plans to deploy approximately 1300 satellites by the end of 2029. The overall GW Constellation Plan comprise two major sub constellations, GW-A59 and GW-2/GW-A2, with a total of 12,992 LEO satellites. These satellites will operate at orbit altitudes of approximately 500–600 km and 1145 km.

The GW constellation simulator is a specialized hardware-in-the-loop testing platform that generates realistic GW satellite signals. It incorporates high-fidelity orbital mechanics models and signal propagation characteristics, with configurable parameters such as satellite constellation, signal power, and measurement noise. Although a public datasheet is not available, the simulator’s performance has been rigorously validated through extensive testing and comparison with theoretical models. In this study, the simulated GW constellation comprises 276 LEO satellites in near-circular orbits at an altitude of approximately 1100 km, ensuring at least 10 satellites are visible to the receiver at any given time. The simulator accounts for relative motion between the satellites and the receiver to compute theoretical Doppler shifts.

The propagation environment was modeled as free space, with additive white Gaussian noise applied to Doppler, position, and velocity measurements to simulate realistic observation errors. The noise amplitudes were set to 0.1 m for satellite position, 0.001 m/s for satellite velocity, and 0.001 Hz for Doppler frequency shift, consistent with typical LEO satellite measurement accuracies [25,26,27,28,29,30,31]. In subsequent research, Doppler observations will be collected by employing a receiver to track satellite signals broadcast from either a simulator or actual satellites in real time.

The signal simulator display interface and zenith map are shown in Figure 2 and Figure 3.

Figure 3 displays the zenith map of a low-orbit constellation, where the purple dots indicate GW low-orbit satellites.

The simulated GW constellation comprised 276 satellites, ensuring that a minimum of 10 satellites were visible to the receiver at any given time, thus satisfying the observational requirements for solving the system of equations. The dataset consisted of observation files and broadcast ephemeris. The observation files contained the measured Doppler shift of the received signal, while the ephemeris files provided the real-time satellite positions and velocities necessary for the positioning calculation.

The starting time of the simulation is 19:02:42, 1 June 2024.

## 4. The Advantages of the Decoupled Algorithm

### 4.1. Limitations of Traditional Method

During a cold start scenario, the receiver’s initial position is typically set to the center of the Earth (0, 0, 0). Although the traditional algorithm can achieve convergence from this point under certain conditions, its performance is not consistently reliable. Figure 4 shows the statistical distribution of positioning errors when the receiver is stationary or moving at a constant velocity (e.g., 100 m/s) along a straight line, starting from the origin (0, 0, 0).

The results in Figure 4 indicate that when the receiver starts from the origin, the traditional algorithm can achieve positioning errors below 10 m for both stationary and dynamic cases.

However, the traditional algorithm fails to converge from other initial points. For instance, when the initial position is set to (−60, −30, 0), the algorithm produces significant errors. Figure 5 presents the problematic positioning results for a stationary receiver and a receiver in uniform linear motion (100 m/s) from this starting point.

As shown in Figure 5a, the stationary receiver’s position solution exhibits a large outlier, causing the traditional algorithm to fail. For the dynamic receiver (Figure 5b), the algorithm produces positioning errors on the order of millions of meters. These results demonstrate that the traditional algorithm cannot guarantee a successful cold start from arbitrary locations. This instability arises from the multiplicative coupling between the receiver’s position and velocity in the system model, where minor deviations can lead to significant errors and divergence in the iterative solution.

### 4.2. Performance of the Proposed Method

The proposed algorithm was tested under the same conditions where the traditional method failed. Figure 6 shows the positioning error when the receiver is initially located at (−60, −30, 0), for both stationary and dynamic (100 m/s) scenarios.

Figure 6 indicates that the positioning error of the proposed algorithm rapidly converges to zero within a short period, regardless of whether the receiver is in a stationary state (Figure 6a) or in a uniform linear motion state (Figure 6b).

Figure 4 demonstrates that the traditional algorithm can converge when initialized at the Earth’s center (0,0,0). However, as shown in Figure 5, it fails drastically when the initial position is set to (−60,−30,0). In contrast, the proposed method successfully handles this challenging initial condition. A comparison between Figure 6 and Figure 5 reveals that our algorithm provides an accurate and rapidly converging solution, with the positioning error approaching zero, whereas the traditional method diverges or produces massive errors. This contrast underscores the traditional algorithm’s critical sensitivity to initial position—a limitation that the proposed method effectively overcomes.

## 5. Performance Under Different Forms of Motion

### 5.1. Experimental Scenarios Design

To comprehensively evaluate the performance of the proposed algorithm, we designed dynamic scenarios with varying motion parameters to simulate real-world applications.

High-speed scenarios (hundreds of m/s): Simulated an aircraft with a trajectory radius exceeding 10,000 m.

Medium-speed scenarios (tens of m/s): Designed two scenarios: a car with a 200 m radius and a train with a kilometer-scale (1000 m) movement radius.

Three-dimensional motion verification: A helicopter scenario was simulated with a spiral ascent motion, matching the train’s speed but incorporating vertical movement.

The specific parameters for these four scenarios are summarized in Table 1.

The corresponding trajectories for these scenarios are illustrated in Figure 7. Due to the large radius in Scenario 1, the trajectory does not form a complete circle within the 5-min simulation period.

We conducted two sets of experiments: one using ideal noise-free signals to validate the algorithm’s fundamental correctness, and another incorporating realistic noise to assess its practical applicability. The noise components, modeled as additive white Gaussian noise, were applied to the Doppler shift, satellite position, and satellite velocity. The real-time LEO orbit accuracy in three directions remains below 0.1 m while velocity accuracy in three directions stays under 0.1 cm/s [25,26,27,28,29,30,31]. Therefore, the amplitude of the satellite position noise was set to 0.1 m and the amplitude of the satellite velocity noise to 0.001 m/s. Assuming that the accuracy of the Doppler frequency shift was approximately 0.001 Hz, the amplitude of frequency noise was set accordingly. In all experiments, the receiver’s initial position, velocity, and clock offset rate were set to zero.

### 5.2. Experiment Result Under Ideal Conditions

Under ideal (noise-free) conditions, the proposed algorithm achieves high precision. Figure 8 shows the 3D position and velocity errors for all four scenarios, demonstrating rapid convergence from the initial state.

Despite significant initial errors, the algorithm’s estimates converge rapidly to the true values within approximately 10 s. For clearer visualization of the steady-state performance, data was resampled at a 5-s interval after the initial convergence. The results are shown in Figure 9.

The resampled data confirms that the position error remains below 1 m and the velocity error below 0.005 m/s during stable tracking. Analysis using 95% probability of error yielded precise 3D position and velocity accuracy presented in Table 2. After taking the average of the number of iterations at all times, the data is listed in the table.

These results demonstrate that the proposed algorithm achieves sub-meter positioning and millimeter-per-second velocity accuracy under ideal conditions, with a computationally feasible iteration count. This accuracy can satisfy the positioning requirements in general situations. The number of iterations for the algorithm is within one hundred.

### 5.3. Experiment Result Under Noise Conditions

In a more realistic scenario with measurement noise, the performance was re-evaluated. The resampled position and velocity errors are shown in Figure 10.

As expected, the introduction of noise increases the error magnitude and introduces higher variability in the results. Analysis using 95% probability of error provided precise 3D position and velocity accuracy presented in Table 3.

The results indicate that the algorithm maintains meter-level positioning accuracy (better than 2 m) and centimeter-per-second velocity accuracy (better than 0.01 m/s) even in the presence of realistic noise. The number of iterations remains manageable, demonstrating the algorithm’s computational efficiency and practical robustness. 

## 6. Performance in High-Speed Scenarios

Certain operational scenarios, particularly in aerospace applications, require receivers to function at very high velocities. To validate the algorithm’s performance under such conditions, we conducted a series of high-speed tests using the simulator. The receiver’s initial position was fixed at (120° E, 50° N, 0 m altitude), and uniform linear motion was simulated with eastward velocities of 1000 m/s, 2000 m/s, 3000 m/s, 4000 m/s, and 5000 m/s, confining the trajectory to the horizontal plane.

The experiments were conducted in a noisy environment, using the same noise parameters as the previous tests. The resulting positioning and velocity accuracies, quantified by the 95% percentile error, are summarized in Table 4. The table also lists the average number of iterations per epoch.

As shown in Table 4, the algorithm’s accuracy does not vary significantly with the increase in receiver velocity. In the presence of noise, it consistently maintains a positioning accuracy of approximately 2 m and a velocity accuracy better than 1 cm/s. The number of iterations remains stable, demonstrating the algorithm’s robustness and computational efficiency across a wide range of high-speed scenarios.

## 7. Performance Across Different Spatial Coordinates

### 7.1. Experimental Design

To evaluate the positioning performance of the algorithm across varying places, 30 points were selected on the Earth’s surface. The selection included meridians at intervals of 60° (0° E, 60° E, 120° E, 180° E, 120° W, and 60° W) and parallel meridians at intervals of 30° (60° N, 30° N, 0° N, 30° S, and 60° S). Six meridians and five parallels were obtained. They intersected at 30 points. Therefore, simulation experiments were conducted at 32 points separately (additional including the North Pole and South Pole).

The receiver followed a spiral upward motion trajectory, with coordinate points positioned at the center of the motion trajectory. To maintain satellite visibility of at least seven satellites, the receiver altitude was limited to 10,000 m during the spiral upward movement. Consequently, this simulation utilized data with a duration of 10 min.

Two scenarios were designed for the experiment, corresponding to different receiver speeds and trajectory radii. The receiver parameters are presented in Table 5.

### 7.2. Result with the GW Constellation

The observed data were processed using a post-processing program for positioning. The 95% percentile 3D position and velocity errors for all 32 points were calculated and are visualized in Figure 11 (for 10 m/s) and Figure 12 (for 1000 m/s).

The results demonstrate that the algorithm achieves meter-level positioning (errors below 2 m) and centimeter-per-second velocity accuracy (errors below 0.02 m/s) across all tested locations. Analysis of the results at different speeds indicates that higher receiver velocities lead to greater fluctuations in positioning error between adjacent points, while the velocity accuracy remains consistently high. This is likely because higher speeds cause more rapid changes in the satellite–receiver geometry, amplifying the subtle differences in satellite visibility across various locations.

These findings confirm that the proposed algorithm can provide reliable positioning and velocity estimation from a zero initial state anywhere on the globe.

### 7.3. Performance with Other Satellite Systems

To further verify the algorithm’s generalizability, it was applied to process simulated signals from the BeiDou Navigation Satellite System. The positioning and velocity results for the 32 global points are shown in Figure 13 (10 m/s) and Figure 14 (1000 m/s).

The results show that while the BeiDou system provides velocity accuracy on the same order of magnitude as the GW system (centimeter-per-second level), its positioning error is significantly larger, on the order of several tens of meters. This performance difference highlights the critical importance of constellation-specific geometry and signal characteristics on the ultimate positioning performance, even when using the same underlying algorithm.

## 8. Conclusions

This study has introduced a robust Doppler positioning algorithm designed for dynamic receivers with unknown initial states by strategically decoupling the traditionally coupled estimation of position and velocity. This core innovation effectively mitigates the convergence issues that plague conventional least-squares methods under large initial uncertainties.

Comprehensive simulations under various dynamic scenarios—including high-speed motion and complex 3D trajectories—validate the algorithm’s performance. The results demonstrate that the proposed method achieves meter-level positioning accuracy and decimeter-per-second velocity accuracy, even in the presence of realistic measurement noise. Crucially, the algorithm accomplishes this from a zero initial state, a condition where traditional algorithms often fail to converge. Furthermore, tests across global coordinates and with different satellite constellations (GW and BDS) confirm the algorithm’s broad applicability, while also highlighting the significant influence of constellation geometry on ultimate positioning performance.

The proposed algorithm establishes a critical foundation for practical LEO-based navigation in GNSS-challenged environments. Although the proposed algorithm demonstrates strong performance in simulations, its real-world deployment depends on the availability of certain system parameters—such as ephemeris format and signal parameters—from LEO constellation operators. As these systems increasingly open their interfaces, the practical implementation of LEO-based positioning is expected to become more feasible. Future work will prioritize validation using real LEO satellite signals. Additional efforts will focus on evaluating the impact of real-world factors such as multipath effects, atmospheric delays, and non-line-of-sight conditions, particularly in urban environments, to further enhance the algorithm’s practical applicability. We will also focus on optimizing the iterative process for resource-constrained devices and exploring deep integration with other sensors (e.g., IMUs) and multi-system fusion (e.g., LEO + GNSS) to enhance robustness and reliability in challenging operational environments.

## Figures and Tables

**Figure 1 sensors-25-06760-f001:**
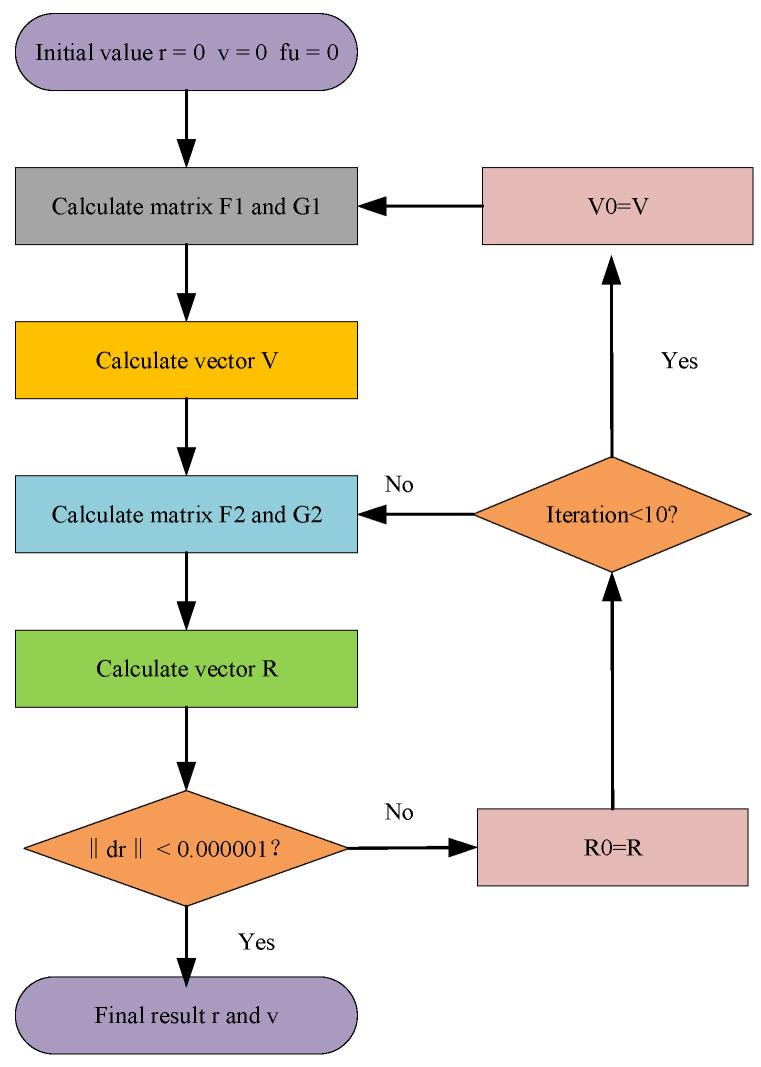
The workflow of the proposed algorithm.

**Figure 2 sensors-25-06760-f002:**
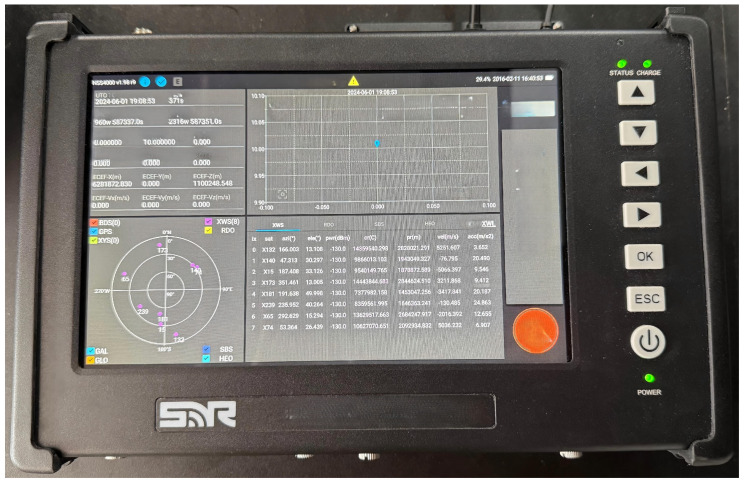
Signal simulator interface.

**Figure 3 sensors-25-06760-f003:**
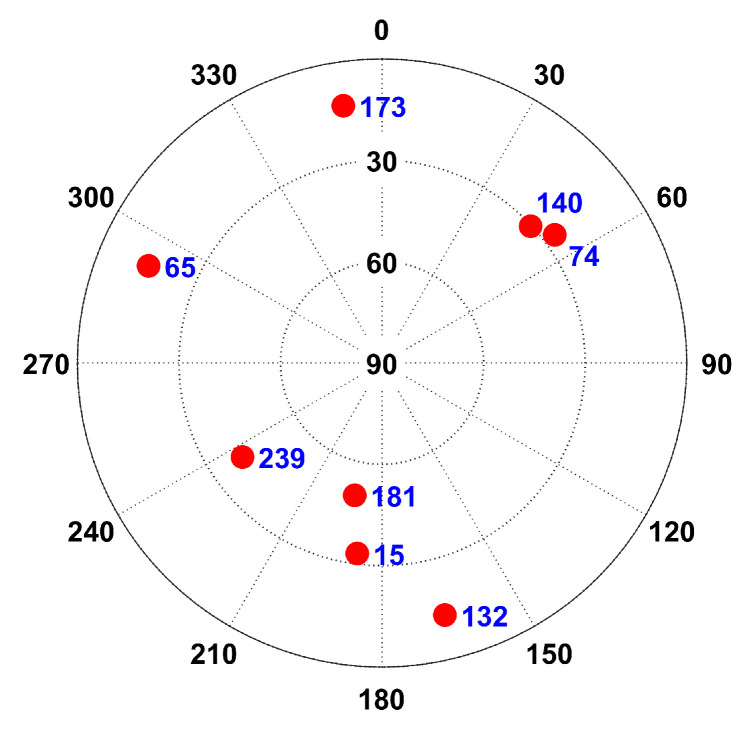
Signal simulator zenith map.

**Figure 4 sensors-25-06760-f004:**
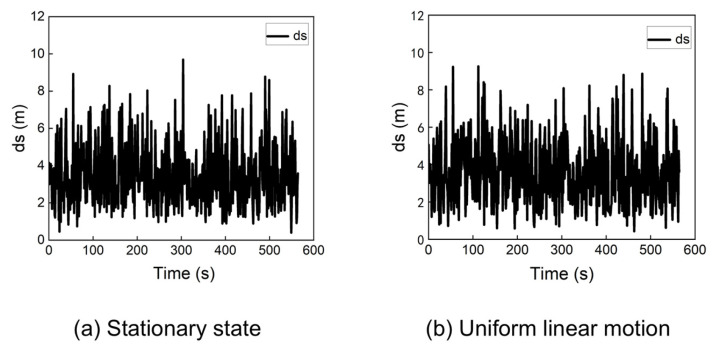
Positioning error of the traditional algorithm at (0, 0, 0) in a stationary state and uniform linear motion.

**Figure 5 sensors-25-06760-f005:**
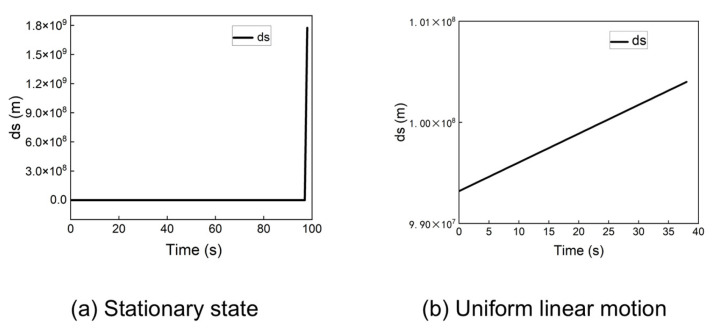
Positioning error of the traditional algorithm at (−60, −30, 0) in a stationary state and uniform linear motion.

**Figure 6 sensors-25-06760-f006:**
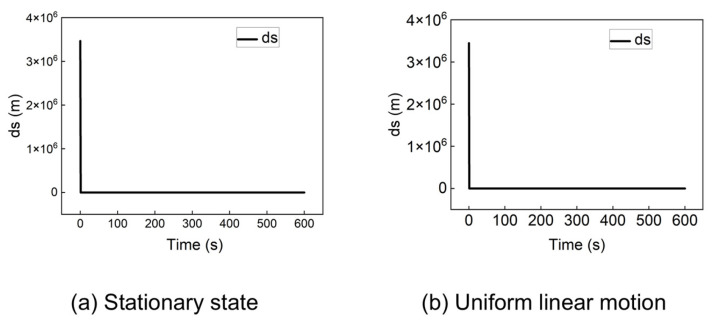
Positioning error of the improved algorithm at (−60, −30, 0) in a stationary state and uniform linear motion.

**Figure 7 sensors-25-06760-f007:**
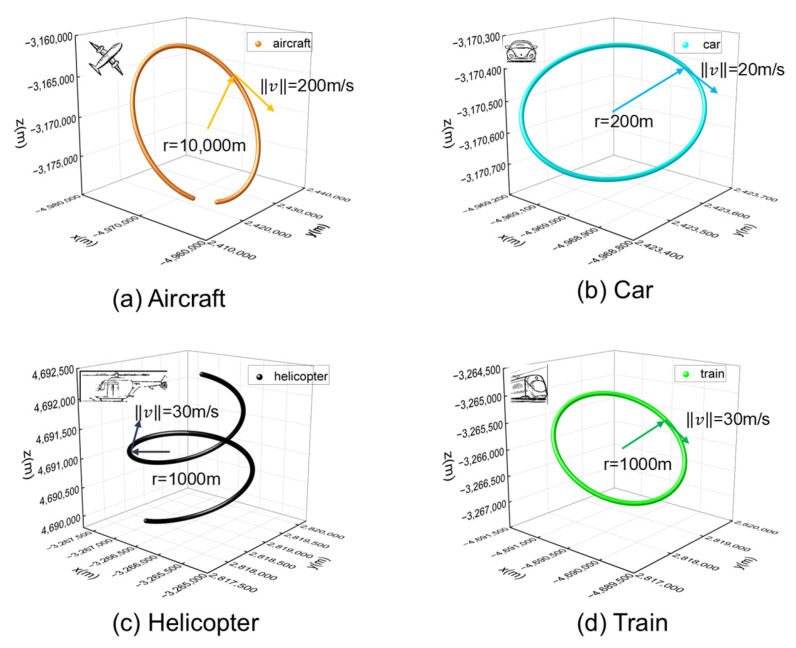
Real trajectory of the four dynamic scenarios.

**Figure 8 sensors-25-06760-f008:**
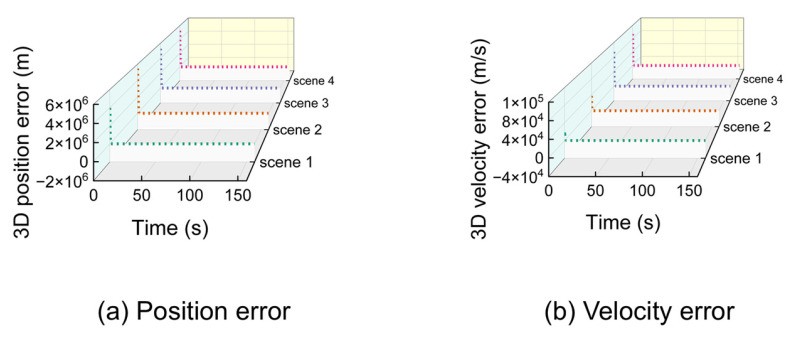
Three-dimensional position and velocity error of 4 scenarios from time 0 under ideal conditions.

**Figure 9 sensors-25-06760-f009:**
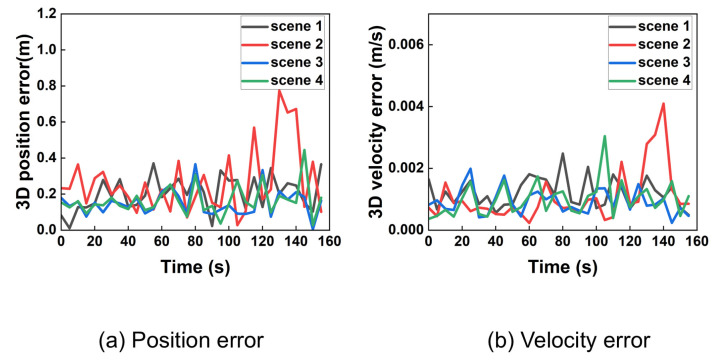
Three-dimensional position and velocity error in ideal condition after extracted.

**Figure 10 sensors-25-06760-f010:**
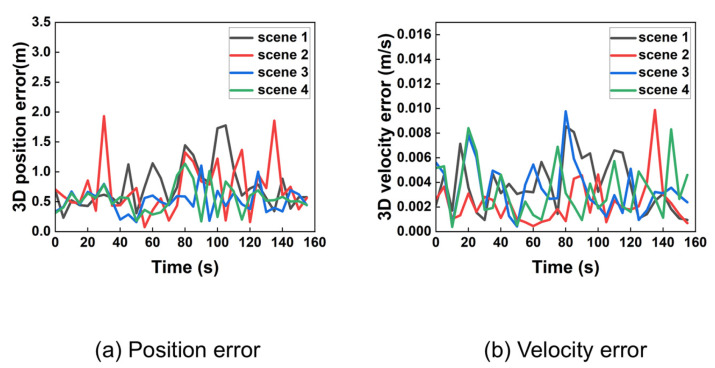
Three-dimensional position and velocity error in noise condition after extracted.

**Figure 11 sensors-25-06760-f011:**
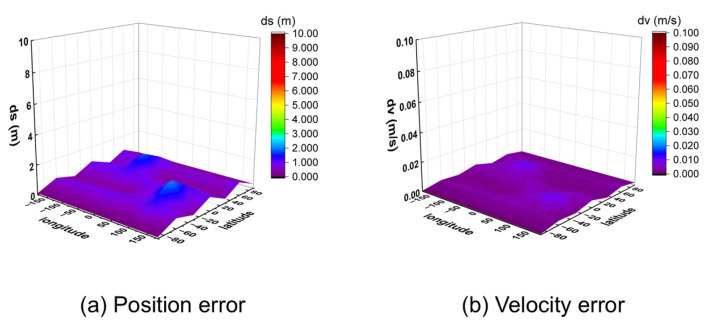
Three-dimensional position and velocity accuracy of 32 coordinate points under spiral upward motion (10 m/s) using GW signal.

**Figure 12 sensors-25-06760-f012:**
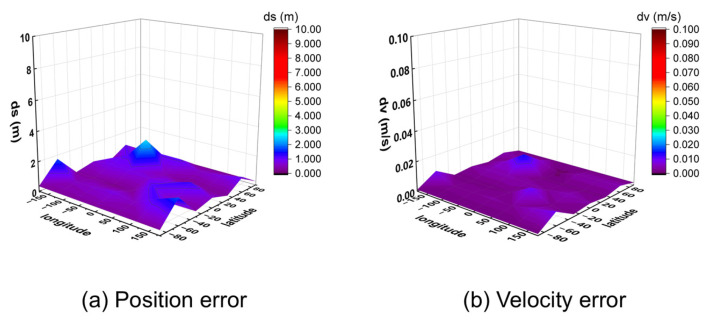
Three-dimensional position and velocity accuracy of 32 coordinate points under spiral upward motion (1000 m/s) using GW signal.

**Figure 13 sensors-25-06760-f013:**
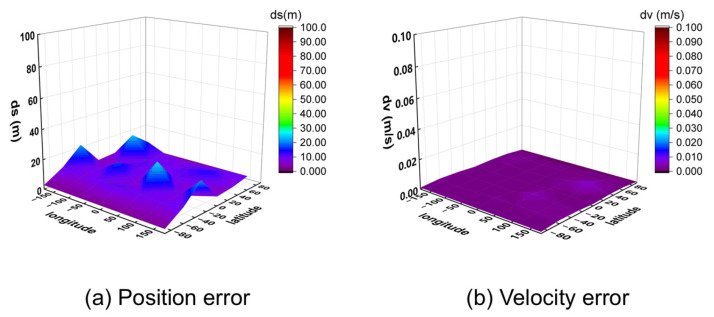
Three-dimensional position and velocity accuracy of 32 coordinate points under spiral upward motion (10 m/s) using BeiDou signal.

**Figure 14 sensors-25-06760-f014:**
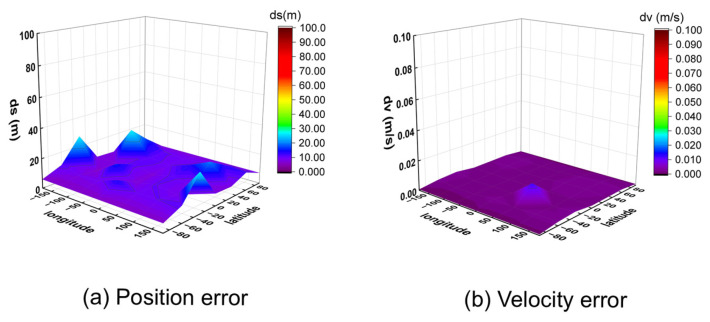
Three-dimensional position and velocity accuracy of 32 coordinate points under spiral upward motion (1000 m/s) using BeiDou signal.

**Table 1 sensors-25-06760-t001:** Motion-model parameters for dynamic receiver.

Number	Carrier	Movement Form	Radius (m)	Velocity (m/s)
1	Aircraft	Circular motion	10,000	200
2	Car	Circular motion	200	20
3	Helicopter	Spiral upward	1000	30
4	Train	Circular motion	1000	30

**Table 2 sensors-25-06760-t002:** Three-dimensional position and velocity accuracy of 4 scenarios under ideal conditions.

Scene	Carrier	3D Position Accuracy (m)	3D Velocity Accuracy (m/s)	Average Number of Iterations
1	Aircraft	0.421730	0.001843	69.758
2	Car	0.530655	0.002089	69.858
3	Helicopter	0.332415	0.001863	39.377
4	Train	0.335265	0.001835	58.829

**Table 3 sensors-25-06760-t003:** Three-dimensional position and velocity accuracy of 4 scenarios under noise condition.

Scene	Carrier	3D Position Accuracy (m)	3D Velocity Accuracy (m/s)	Average Number of Iterations
1	Aircraft	1.73801032	0.00745809	41.411
2	Car	1.92943429	0.00740109	40.8
3	Helicopter	1.33587983	0.00682125	45.563
4	Train	1.29767126	0.00675676	48.787

**Table 4 sensors-25-06760-t004:** Three-dimensional position and velocity accuracy at different velocities under noise condition.

Number	Speed (m/s)	3D Position Accuracy (m)	3D Velocity Accuracy (m/s)	Average Number of Iterations
1	1000 m/s	1.62612352	0.00711216	69.745
2	2000 m/s	1.44138400	0.00612194	70.011
3	3000 m/s	1.33264933	0.00470396	71.078
4	4000 m/s	1.31514958	0.00447492	65.752
5	5000 m/s	1.45109335	0.00457131	67.275

**Table 5 sensors-25-06760-t005:** Motion-model parameters for spiral upward motion.

Number	Movement Form	Radius (m)	Velocity (m/s)
1	Uniform spiral motion	100	10
2	Uniform spiral motion	10,000	1000

## Data Availability

The original contributions presented in this study are included in the article. Further inquiries can be directed to the corresponding authors.

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
