# Peer review of "A Decoupled Doppler Positioning Algorithm for Dynamic Receivers Using LEO Constellation Signals"

_sensors, 2025, doi:10.3390/s25216760_

Round 1

Reviewer 1 Report

Comments and Suggestions for Authors

The manuscript entitled “A Decoupled Doppler Positioning Algorithm for Dynamic Receivers Using LEO Constellation Signals” presents an enhanced least-squares-based Doppler positioning algorithm for dynamic receivers utilizing signals from low Earth orbit (LEO) communication constellations.

The authors propose a novel approach that decouples the estimation of position and velocity, thereby enabling robust positioning from a zero initial state and effectively mitigating convergence issues commonly encountered in conventional least-squares methods under large initial uncertainties. Simulation results demonstrate that the proposed algorithm achieves meter-level positioning accuracy and decimeter-per-second velocity accuracy under various dynamic scenarios, including high-speed motion. Additionally, the authors provide comparative evaluations using different satellite constellations (XWS and BeiDou) and across global coordinate frames, which further enhance the credibility and applicability of their approach.

Overall, the paper is well-structured, with clear writing and compelling results. The proposed method demonstrates strong potential for real-world deployment. I consider the work to be suitable for publication.

However, from a practical application standpoint, the use of LEO satellites for positioning requires cooperation from system providers, particularly in terms of access to ephemeris and other auxiliary data. Therefore, while the proposed algorithm shows promise, its deployment in real-world scenarios may still take time.

As an exploratory and early-stage study, this work is valuable and merits publication. Nonetheless, future research should consider additional real-world factors such as multipath effects at the receiver end, which can significantly impact positioning accuracy, especially in urban or complex environments. I look forward to seeing continued advancements in this area.

With the rapid development of LEO satellite communications, the use of LEO satellites for Positioning, Navigation, and Timing (PNT) has become an increasingly important and timely topic. This technology has the potential to complement traditional GNSS systems and enhance overall system resilience, opening up new avenues for application development. Many readers will likely be interested in this emerging area. In this context, Doppler-based positioning algorithms represent a key component of PNT innovation. The study contributes to this field by proposing an improved algorithm that enhances the convergence speed and accuracy of position and velocity estimation compared to traditional methods. This advancement is both relevant and valuable, and it aligns well with current trends in satellite-based navigation research.   I am generally satisfied with the manuscript and believe it is suitable for publication. The only concern is that the authors have provided simulation results only, which limits the extent to which I can evaluate the study. For example, I have one concern regarding the simulation data and experimental design. The authors stated that the simulation dataset consisted of observation files and broadcast ephemeris. However, in the subsequent section on Experimental Scenarios Design, they describe scenarios involving high-speed motion and the selection of 30 points on the Earth’s surface. In practice, it would be difficult to obtain observation files that contain measured Doppler shifts of received signals under such conditions. Therefore, I would appreciate clarification from the authors on how the Doppler shift data were generated or simulated in these scenarios. Providing more details on the simulation methodology would help clarify the applicability of the proposed algorithm. 

Author Response

Issue 1:

From a practical application standpoint, the use of LEO satellites for positioning requires cooperation from system providers, particularly in terms of access to ephemeris and other auxiliary data. Therefore, while the proposed algorithm shows promise, its deployment in real-world scenarios may still take time.

Response 1:

We fully agree with the reviewer regarding the practical challenges of accessing real-time ephemeris and signal specifications from LEO constellation operators. Our work primarily focuses on establishing a robust algorithmic foundation for Doppler-based positioning under dynamic conditions with zero initial state. As LEO constellations (e.g., Starlink, OneWeb) gradually open their interfaces to third-party developers, we believe our method provides a timely and viable framework for future real-world implementations. We have added a discussion on this aspect in the Conclusion section.

Correction 1:

Added a discussion on practical deployment challenges in the conclusion section: "Although the proposed algorithm demonstrates strong performance in simulations, its real-world deployment depends on the availability of certain system parameters—such as ephemeris format and signal parameters — from LEO constellation operators. As these systems increasingly open their interfaces, the practical implementation of LEO-based positioning is expected to become more feasible. "

(l.413-418)

Issue 2:

Future research should consider additional real-world factors such as multipath effects at the receiver end, which can significantly impact positioning accuracy, especially in urban or complex environments.

Response 2:

We thank the reviewer for this valuable suggestion. In this study, we focused on validating the convergence and initial-state robustness of the proposed decoupled algorithm. The inclusion of multipath effects and other channel impairments—particularly in urban canyon environments—is indeed an important direction for future research. We have added a statement in the Conclusion section outlining our plans to address these factors in subsequent work.

Correction 2:

Add in future work in the conclusion section: "Additional efforts will focus on evaluating the impact of real-world factors such as multipath effects, atmospheric delays, and non-line-of-sight conditions, particularly in urban environments, to further enhance the algorithm's practical applicability."

(l.418-421)

Issue 3:

I would appreciate clarification from the authors on how the Doppler shift data were generated or simulated in these scenarios. Providing more details on the simulation methodology would help clarify the applicability of the proposed algorithm.

Response 3:

We have expanded Section 3 (Simulation Data Sources) to provide a more detailed description of the Doppler shift data generation process. The data were generated using the XWS constellation simulator provided by Hunan Weidao Information Technology Co., Ltd., which models satellite orbits and velocities based on broadcast ephemeris and generates realistic Doppler observations including additive white Gaussian noise.

Correction 3:

Add a detailed description of the simulation method in Section 3: "The simulator accounts for relative motion between the satellites and the receiver to compute theoretical Doppler shifts.

The propagation environment was modeled as free space, with additive white Gaussian noise applied to Doppler, position, and velocity measurements to simulate realistic observation errors. The noise amplitudes were set to 0.1 m for satellite position, 0.001 m/s for satellite velocity, and 0.001 Hz for Doppler frequency shift, consistent with typical LEO satellite measurement accuracies [25-31]."

(l.190-196)

Reviewer 2 Report

Comments and Suggestions for Authors

In this manuscript, the authors propose a doppler positioning algorithm which is built for LEO constellation signals. The approach is able to decouple the position and the velocity estimations, and it is validated through numerical simulations using a constellation simulator device. The manuscript is interesting and tackles an interesting topic in the field of LEO PNT, which is a very hot topic in the current scientific and industrial literature. Nevertheless, it is necessary to implement some improvements to make the manuscript ready for publication. Here it is a summary.

l.29: Once the acronym has been proposed as extended the first time, you can avoid to repeat it.

l.35-37: Since the paper is written in 2025, maybe you can update this information and correctly reference it.

l.36-44: I think this part can be expanded and it should be given more emphasis proposing a more robust taxonomy of the different uses of LEO satellites, for communications and PNT applications.

l.57-58: You can also take into account recent works which are based on signals of opportunity, in particular those systems which provide LEO coarse position estimation systems based on angle of arrival estimation rather than doppler estimation.

l.165: Can you put a reference to the datasheet of the instrument?

l.168: Always indicate the figures explicitly by their numbers.

l.169: Please improve the image quality or put a screenshot.

l.174: Which constellation are you targeting? If you are not targeting a specific constellation, please provide the details of the orbits and on the simulation setting. Also, please provide the information on the propagation environment.

l.216: It is not clear what is the difference between the two figures. Please better highlight the differences.

Author Response

Issue 1:

l.29: Once the acronym has been proposed as extended the first time, you can avoid to repeat it.

Response 1:

We have revised the manuscript to remove redundant acronym expansions after their first occurrence throughout the text.

Correction 1:

We conducted a full-text check and removed duplicate abbreviation expansions to ensure that each abbreviation is only expanded when it first appears.

(l.29)

Issue 2:

l.35-37: Since the paper is written in 2025, maybe you can update this information and correctly reference it.

Response 2:

We have updated the in-orbit LEO satellite count to reflect the latest available data and added corresponding references.

Correction 2:

We have updated the description of the number of satellites in the introduction section: " As of August 2025, the number of in-orbit LEO satellites has exceeded 10000, solidifying LEO-based technologies as a primary focus in navigation research."

(l.35-36)

Issue 3:

l.36-44: I think this part can be expanded and it should be given more emphasis proposing a more robust taxonomy of the different uses of LEO satellites, for communications and PNT applications.

Response 3:

We have expanded the introduction to include a more comprehensive taxonomy of LEO satellite applications, and categorize their use in communications, Earth observation, and PNT services.

Correction 3:

We add a paragraph in the introduction about the classification of LEO satellite applications: "LEO satellites support a wide variety of applications beyond communications, including Earth observation, weather monitoring, and Positioning, Navigation, and Timing (PNT) services. In the field of PNT, LEO constellations can enhance traditional GNSS systems by providing stronger signals, faster convergence, and improved robustness in challenging environments [9-13]."

(l.38-42)

Issue 4:

l.57-58: You can also take into account recent works which are based on signals of opportunity, in particular those systems which provide LEO coarse position estimation systems based on angle of arrival estimation rather than doppler estimation.

Response 4:

We have added a discussion on recent LEO positioning methods based on Angle-of-Arrival (AoA) estimation and included relevant references in the Introduction.

Correction 4:

We supplement the relevant research on AoA method in the introduction: " Besides Doppler-based methods, recent studies have explored Angle-of-Arrival (AoA) estimation using LEO signals of opportunity for coarse positioning [22-23]. While AoA approaches offer complementary benefits, this work focuses on Doppler-based positioning due to its inherent robustness to frequency offsets and compatibility with existing communication infrastructure. "

And add corresponding references: “22. Guo, J.Q.; Wang, Y. Efficient AOA estimation and NLOS signal utilization for LEO constellation-based positioning using satellite ephemeris information. Applied sciences-basel, 2025, 15(3).” “23. Florio, A.; Bnilam, N.; Talarico, C.; Crosta, P.; Avitabile, G.; Coviello, G. LEO-Based Coarse Positioning Through Angle-of-Arrival Estimation of Signals of Opportunity. IEEE Access, 2024, 12:17446-17459.”

(l.61-65)

Issue 5:

T l.165: Can you put a reference to the datasheet of the instrument?

Response 5:

The simulator is a proprietary tool developed by our industrial partner, and a public datasheet is not available. However, we have added a detailed description of its simulation principles and parameters in Section 3.

Correction 5:

We add a detailed description of the simulator in Section 3: " The GW constellation simulator is a specialized hardware-in-the-loop testing platform that generates realistic GW satellite signals. It incorporates high-fidelity orbital mechanics models and signal propagation characteristics, with configurable parameters such as satellite constellation, signal power, and measurement noise. Although a public datasheet is not available, the simulator's performance has been rigorously validated through extensive testing and comparison with theoretical models. In this study, the simulated GW constellation comprises 276 LEO satellites in near-circular orbits at an altitude of approximately 1,100 km, ensuring at least 10 satellites are visible to the receiver at any given time. The simulator accounts for relative motion between the satellites and the receiver to compute theoretical Doppler shifts.”

(l.182-191)

Issue 6:

l.168: Always indicate the figures explicitly by their numbers.

Response 6:

We have carefully reviewed the manuscript to ensure all figures and tables are explicitly referenced by their numbers.

Correction 6:

We conducted a full-text check and corrected all chart references to ensure that each chart was clearly numbered and referenced when first mentioned in the text.

(l.94, l.141, l.200)

Issue 7:

l.169: Please improve the image quality or put a screenshot.

Response 7:

We have replaced Figures 2 and 3 with higher-resolution versions to improve readability.

Correction 7:

We regenerate Figure 2 and Figure 3 using higher resolution images to ensure that all text and details are clear and distinguishable.

(l.202, l.206)

Figure 2. Signal simulator interface.

Figure 3. Signal simulator zenith map.

Issue 8:

l.174: Which constellation are you targeting? If you are not targeting a specific constellation, please provide the details of the orbits and on the simulation setting. Also, please provide the information on the propagation environment.

Response 8:

We are targeting the Chinese GW constellation. In the simulator, its code name is XWS. We have changed it to the more well-known name GW. In this article revision, we have clarified that we used the simulated GW constellation and added details about the orbital parameters and propagation environment in Section 3.

Correction 8:

We add constellation and propagation environment details in Section 3: " The GW constellation is a Chinese LEO satellite system currently under construction. To date, 116 satellites have been launched, with plans to deploy approximately 1300 satellites by the end of 2029. The overall GW Constellation Plan comprise two major sub constellations, GW-A59 and GW-2/GW-A2, with a total of 12992 LEO satellites. These satellites will operate at orbit altitudes of approximately 500-600km and 1145km. The GW constellation simulator is a specialized hardware-in-the-loop testing platform that generates realistic GW satellite signals. It incorporates high-fidelity orbital mechanics models and signal propagation characteristics, with configurable parameters such as satellite constellation, signal power, and measurement noise. Although a public datasheet is not available, the simulator's performance has been rigorously validated through extensive testing and comparison with theoretical models. In this study, the simulated GW constellation comprises 276 LEO satellites in near-circular orbits at an altitude of approximately 1,100 km, ensuring at least 10 satellites are visible to the receiver at any given time."

(l.177-190)

Issue 9:

l.216: It is not clear what is the difference between the two figures. Please better highlight the differences.

Response 9:

The two images are very similar, with the difference being that they correspond to two different forms of motion. The Figure 6(a) represents a stationary state, while the Figure 6(b) represents uniform linear motion. The existence of Figure 6 is for comparison with Figure 5. So, they correspond structurally. From the results, the localization results of the two forms of motion are similar, which is why there is not much difference between the two graphs (a) and (b) in Figure 6. We have revised the caption and text in Section 4.1 to explicitly contrast the convergence behavior of the traditional algorithm from different initial positions.

Correction 9:

We clearly explain the difference between Figure 4 and Figure 5 and Figure 6 in Section 4.1: "Figure 6 indicates that the positioning error of the proposed algorithm rapidly converges to zero within a short period, regardless of whether the receiver is in a stationary state (Figure 6(a)) or in a uniform linear motion state (Figure 6(b)). Figure 4 demonstrates that the traditional algorithm can converge when initialized at the Earth's center (0, 0, 0). However, as shown in Figure 5, it fails drastically when the initial position is set to (-60, -30, 0). In contrast, the proposed method successfully handles this challenging initial condition. A comparison between Figure 6 and Figure 5 reveals that our algorithm provides an accurate and rapidly converging solution, with the positioning error approaching zero, whereas the traditional method diverges or produces massive errors. This contrast underscores the traditional algorithm's critical sensitivity to initial position—a limitation that the proposed method effectively overcomes."

(l.255-265)

Round 2

Reviewer 2 Report

Comments and Suggestions for Authors

The authors addressed all my comments in a comprehensive way. Therefore, I can now recommend the paper for publication in this form. Good luck with the rest of the review process.